# Colonization with multi-drug-resistant organisms negatively impacts survival in patients with non-small cell lung cancer

Jan A. Stratmann[1]*, Raphael Lacko[1], Olivier Ballo[1], Shabnam Shaid[1],
Wolfgang Gleiber[2], Maria J. G. T. Vehreschild[3,4], Thomas Wichelhaus[4,5,6],
Claudia Reinheimer[4,5,6], Stephan Göttig[4,5], Volkhard A. J. Kempf[4,5,6], Peter Kleine[7],
Susanne Stera[8], Christian Brandts[1,9], Martin Sebastian[1], Sebastian Koschade[1]

1 Department of Internal Medicine, Hematology/Oncology, Goethe University, Frankfurt, Frankfurt am Main,
Germany, 2 Department of Internal Medicine, Pneumology, Goethe University, Frankfurt, Frankfurt am Main,
Germany, 3 Department of Internal Medicine, Infectious Diseases, Goethe University, Frankfurt, Frankfurt
am Main, Germany, 4 University Center for Infectious Diseases, Goethe University, Frankfurt, Frankfurt am
Main, Germany, 5 Institute of Medical Microbiology and Infection Control, Goethe University, Frankfurt,
Frankfurt am Main, Germany, 6 University Center of Competence for Infection Control, Frankfurt, State of
Hesse, Germany, 7 Department of Cardiothoracic Surgery, Goethe University, Frankfurt, Frankfurt am Main,
Germany, 8 Department of Radiation Oncology, Goethe University, Frankfurt, Frankfurt am Main, Germany,
9 University Cancer Center Frankfurt (UCT), Goethe University, Frankfurt, Germany

* jan.stratmann@kgu.de

pone.0242544

Health and Allied Sciences School of Medicine,
UNITED REPUBLIC OF TANZANIA

## Abstract

### Objectives

Multidrug-resistant organisms (MDRO) are considered an emerging threat worldwide. Data
covering the clinical impact of MDRO colonization in patients with solid malignancies, how-
ever, is widely missing. We sought to determine the impact of MDRO colonization in patients
who have been diagnosed with Non-small cell lung cancer (NSCLC) who are at known high-
risk for invasive infections.

### Materials and methods

Patients who were screened for MDRO colonization within a 90-day period after NSCLC
diagnosis of all stages were included in this single-center retrospective study.

### Results

Two hundred and ninety-five patients were included of whom 24 patients (8.1%) were
screened positive for MDRO colonization (MDRO^pos) at first diagnosis. *Enterobacterales*
were by far the most frequent MDRO detected with a proportion of 79.2% (19/24). MDRO
colonization was present across all disease stages and more present in patients with con-
comitant diabetes mellitus. Median overall survival was significantly inferior in the MDRO^pos
study group with a median OS of 7.8 months (95% CI, 0.0–19.9 months) compared to a
median OS of 23.9 months (95% CI, 17.6–30.1 months) in the MDRO^neg group in univariate
(p = 0.036) and multivariate analysis (P = 0.02). Exploratory analyses suggest a higher rate

**Data Availability Statement:** All relevant data are within the manuscript and its Supporting Information files.

**Funding:** The authors received no specific funding for this work.

**Competing interests:** I have read the journal's policy and the authors of this manuscript have the following competing interests: JS reports personal fees from Bristol-Myers & Squibb, personal fees from Novartis, personal fees from Roche, outside the submitted work. RL has nothing to disclose. OB has nothing to disclose. SS has nothing to disclose. WG has nothing to disclose. MJGTV has served at the speakers' bureau of Akademie für Infektionsmedizin, Ärztekammer Nordrhein, Astellas Pharma, Basilea, Gilead Sciences, Merck/MSD, Organobalance, Pfizer and Uniklinik Freiburg / Kongress und Kommunikation, received research funding from 3M, Astellas Pharma, DaVolterra, Gilead Sciences, MaaT Pharma, Merck/MSD, Morphochem, Organobalance, Seres Therapeutics, and is a consultant to Alb-Fils Kliniken GmbH, Ardeypharm, Astellas Pharma, Berlin Chemie, DaVolterra, Ferring, MaaT Pharma and Merck/MSD. TW has nothing to disclose. CR has nothing to disclose. SG has nothing to disclose. VK reports grants and personal fees from DFG Germany (DFG FG 2251), grants and personal fees from EU Marie Sklodowska-Curie (#765042), during the conduct of the study. PK has nothing to disclose. SS has nothing to disclose. CB has nothing to disclose. MSe reports personal fees from Lilly, personal fees from Astra-Zeneca, personal fees from Bristol-Myers & Squibb, personal fees from Merck Sharp & Dohme, personal fees from Pfizer, personal fees from Takeda, personal fees from Roche, personal fees from AbbVie, personal fees from Boehringer Ingelheim, personal fees from Celgene, personal fees from Novartis, outside the submitted work. SK has nothing to disclose. This does not alter our adherence to PLOS ONE policies on sharing data and materials. There are no patents, products in development or marketed products associated with this research to declare.

of non-cancer-related-mortality in MDRO[pos] patients compared to MDRO[neg] patients (p = 0.002) with an increased rate of fatal infections in MDRO[pos] patients (p = 0.0002).

## Conclusions

MDRO colonization is an independent risk factor for inferior OS in patients diagnosed with NSCLC due to a higher rate of fatal infections. Empirical antibiotic treatment approaches should cover formerly detected MDR commensals in cases of (suspected) invasive infections.

## Introduction

Multidrug-resistant organisms (MDRO) such as vancomycin-resistant *Enterococci* (VRE), third-generation Cephalosporin-resistant *Enterobacterales*, piperacillin/tazobactam-resistant *Pseudomonas aeruginosa* and *Methicillin-resistant Staphylococcus aureus* (MRSA) are considered an emerging threat worldwide as there are fewer and sometimes even no antimicrobial agents left to treat infections caused by these pathogens [1, 2]. The impact of MDRO in patients with hematologic malignancy has been investigated extensively [3–6]. Hematologic patients colonized with MDRO are at a profound risk of invasive MDRO infections [7–9]. Infections with MDRO provoke prolonged hospital stays, increased hospital costs and negatively impact survival [3, 10–14].

However, only few clinical studies have addressed the impact of MDRO infections (compared to non-MDRO infections) in patients with solid malignancies. As most of these analyses suffer from several limitations such as focusing solely on critically ill patients treated on intensive care units [11, 15, 16], providing only short-term follow-ups [17] or including various oncological entities at different disease stages [18], valid conclusions on the overall survival impact of colonization and infection with MDRO in patients with solid malignancies cannot be drawn.

Non-small cell lung cancer (NSCLC) is the leading cause of cancer-related death worldwide [19]. Most patients are diagnosed in advanced disease stages and palliative treatment choices consist of targeted therapy, immunotherapy and cytotoxic agents such as platinum–based chemotherapy. Patients at all stages are at high risk of life-threatening infections due to invasive therapeutic and diagnostic procedures, immunocompromising therapy and related comorbidities (e.g. chronic obstructive pulmonary disease) [20]. Large prospective clinical trials report bacterial infection rates in approximately 10% of patients with limited or advanced disease stages [21–27] and up to 70% in retrospective analyses [28, 29]

The presence of MDRO colonization in patients with NSCLC and its impact on survival has not been investigated so far. We therefore sought to determine the frequency, clinical characteristics and clinical impact with a focus on survival outcomes of MDRO colonization in patients with NSCLC in a retrospective single center analysis.

## Material and methods

### Defining the study population

Patients diagnosed with NSCLC, stages I-IV according to the Union International Contre le Cancer (UICC) 7th edition between 2012 and 2016 and screened for MDRO (definition see below under "screening procedures and definitions") within a time period of 90 days calculated from pathological confirmed first diagnosis of NSCLC were included to this analysis.

Exclusion criteria were history of or concomitant underlying second malignancy—aside from localized non-melanoma skin cancer (e.g. basalioma) that had been curatively treated -, insufficient case documentation and missing MDRO screening. Patient data used in this study were provided by the University Cancer Center Frankfurt (UCT). Written informed consent was obtained from all patients and the study was approved by the institutional Review Boards of the UCT and the Ethical Committee at the University Hospital Frankfurt (project-number: STO-01-2016, Amendment 1, 06.06.2018).

## Screening procedure and definitions

According to German infection law (Infektionsschutzgesetz, IfSG) [30] execution of an infection control protocol in order to prevent the transmission of infective agents, such as MDRO is mandatorily required. At the University hospital Frankfurt, this legal requirement by IfSG as well as the recommendations of the Commission for Hospital Hygiene and Infection Prevention (KRINKO) at the Robert Koch Institute, Berlin, Germany (e.g. recommendations for prevention and control of MRSA in medical and nursing facilities; [31]) are entirely fulfilled. Therefore, patients reporting defined risk factors, e.g. arriving from high-prevalence countries, e.g. including but not limited to countries from the middle east, south-east Asia and India for MDRO, being refugee as well as patients admitted to any intensive/intermediate care unit as well as all patients admitted to the thoracic surgery ward and patients admitted to the clinical oncology ward need to be screened for MDRO at the day of admittance by nasal, rectal and pharyngeal swabs [32, 33].

MDRO were defined as *Enterococcus faecium* or *Enterococcus faecalis* with vancomycin resistance (VRE) and *Methicillin*-resistant *Staphylococcus aureus* (MRSA). Multidrug-resistant gram-negative bacteria were defined as *Klebsiella pneumoniae*, *Klebsiella oxytoca*, *Escherichia coli*, *Proteus mirabilis* with extended spectrum beta–lactamase (ESBL)–like phenotype as well as *Enterobacterales*, *Acinetobacter baumannii* and *Pseudomonas aeruginosa* resistant against piperacillin, any 3rd/4th generation Cephalosporin, and fluoroquinolones ± carbapenems [31].

Patients were defined as "colonized" if an MDRO was detected (MDRO[pos]) in at least one nasal, rectal or pharyngeal swab. Screened patients without evidence of MDRO colonization were defined as MDRO[neg]. In case of multiple MDRO screenings within the predefined time period at first diagnosis, the first screening result defined group assignment.

## Detection and molecular resistance analysis in MDRO

Rectal swabs were collected using culture swabs with Amies collection and transport medium (Hain Lifescience, Nehren, Germany) and were afterwards streaked onto CHROMagarTM ESBL plates (Mast Diagnostica, Paris, France), chromID CARBA (bioMérieux, Nürtingen, Germany), chromID VRE (bioMérieux), chromID OXA-48 (bioMérieux), Brilliance MRSA-Agar (Oxoid, Wesel, Germany). Matrix-assisted-laser desorption ionization-time of flight analysis (MALDI–TOF) and VITEK2 (bioMérieux) were used to identify gram negative species, when growth was detected. Antibiotic susceptibility testing was carried out according to the Clinical and Laboratory Standards Institute (CLSI) guidelines by VITEK 2 and antibiotic gradient tests (bioMérieux) or agar diffusion (Oxoid). Carbapenemase encoding genes were detected via polymerase chain reaction analysis and subsequent sequencing from carbapenem-resistant *Enterobacterales* including the *bla* genes for carbapenemases OXA–48, OXA–48 like and KPC, NDM, VIM, IMP as well as OXA–23, OXA–24, OXA–51, and OXA– 58 for *A. baumannii* [34]. For the detection of MRSA, nasal and pharyngeal swabs were inoculated on Brilliance MRSA Agar (Oxoid, Wesel, Germany). Identification of MRSA species was done by

MALDI–TOF and antibiotic susceptibility testing using VITEK 2. The clonal identity of MRSA isolates was analyzed by staphylococcal protein A (*spa*) typing using the Ridom Staph-Type software (Ridom GmbH, Würzburg, Germany), as previously reported [32, 34]. All laboratory testing was performed under strict quality-controlled criteria (laboratory accreditation according to ISO 15189:2007 standards; certificate number D–ML–13102–01–00, valid through January 25th, 2021).

## Study endpoints

Predefined primary study endpoints were event-free-survival (EFS) and overall survival (OS) compared between MDRO[pos] and MDRO[neg] groups, taking into account known confounding variables such as gender, age, disease stage, Eastern Cooperative Oncology Group (ECOG) Performance Status, NSCLC histology, smoking status and concomitant diseases in multivariate analysis. Event-free-survival was defined as the time period until re-occurrence of histologically confirmed lung cancer after curative treatment or the time period until next treatment line or death from any cause, whichever came first. Patients who were still alive at data cut-off were censored with regard to OS at the date of last contact. Patients who did not die or did not show any of the above-mentioned events at the time of the data cut-off were censored with regard to EFS analysis at the date of last contact.

Secondary endpoints were the distribution of causes of death stratified by MDRO colonization status and number and length of hospital stays stratified by cause of inpatient treatments. The specific causes of death were extracted from the letter of notification or death certificate. Exploratory endpoints were the rate of subsequently detected invasive MDRO infections and evaluation of antibiotic approaches in MDRO[pos] patients with infectious complications. Finally, we planned to compare the eligible study cohort to patients who were primarily excluded from the analysis due to missing MDRO screening (off-target population).

## Statistical analysis

The number of all included patients and recorded variables were reported descriptively. Survival analyses were performed using the Kaplan-Meier method for estimation of the percentage of surviving patients and the log-rank test for comparing patient groups. Cox proportional hazard regression analysis was used for multivariate analyses. Proportional hazards assumption and residuals were checked formally and graphically. Schoenfeld residuals for all covariates were verified to be independent of time. Competing risks of death and their cumulative incidences were analyzed using R's cmprsk package implementing the proportional subdistribution hazards' regression model described in Fine and Gray (1999) [35] with failure types as indicated and MDRO colonization as a binary covariate. Comparative analyses for differences in proportions and other numerical variables between study groups were performed using Chi$^2$ test and Mann-Whitney U test. A p-value below 0.05 was considered statistically significant. R version 3.5.1 and GraphPad Prism version 6.01 were used for statistical analysis and reporting of the data collected for this study.

## Results

### Study population and off-target analysis

We identified 639 patients diagnosed with NSCLC between 2012 and 2016 in the institutional cancer registry of the University Hospital, Frankfurt am Main, Germany of whom 295 were available for further analysis. A CONSORS flow chart showing the process of inclusion of eligible patients into the analysis is available in S1 Fig in S1 File. Twenty-four patients (8.1%) were

**Table 1. Patient and disease characteristics.**

| | | All patients (n = 295) | No MDRO (n = 271) | MDRO colonization (n = 24) | P value* |
|---|---|---|---|---|---|
| Gender | female | 110 (37%) | 106 (39%) | 4 (17%) | .05 |
| | male | 185 (63%) | 165 (61%) | 20 (83%) | .05 |
| Age at diagnosis, median (range), years | | 67 (29–90) | 67 (29–90) | 70 (53–90) | .17 |
| Smoking history | | 209 (71%) | 192 (71%) | 17 (71%) | 1 |
| ECOG performance score ≤ 2 (%) | | 281 (95%) | 259 (96%) | 22 (92%) | .72 |
| UICC | IA | 50 (17%) | 45 (17%) | 5 (21%) | .81 |
| | IB | 19 (6%) | 18 (7%) | 1 (4%) | .97 |
| | IIA | 22 (7%) | 21 (8%) | 1 (4%) | .81 |
| | IIB | 25 (9%) | 24 (9%) | 1 (4%) | .68 |
| | IIIA | 61 (21%) | 54 (20%) | 7 (29%) | .42 |
| | IIIB | 19 (6%) | 19 (7%) | 0 (0%) | .36 |
| | IV | 99 (34%) | 90 (33%) | 9 (38%) | .84 |
| Presence of brain metastases | | 48 (16%) | 47 (17%) | 1 (4%) | .17 |
| Histology | Adeno NSCLC | 160 (54%) | 150 (55%) | 10 (42%) | .28 |
| | SCNSCLC | 127 (43%) | 113 (42%) | 14 (58%) | .17 |
| | other | 8 (3%) | 8 (3%) | 0 (0%) | .84 |
| Mutations (pos. / neg.) | ALK | 3 (1%) / 32 (11%) | 2 (1%) / 29 (11%) | 1 (4%) / 3 (13%) | |
| | BRAF | 2 (1%) / 5 (2%) | 1 (0%) / 4 (1%) | 1 (4%) / 1 (4%) | |
| | EGFR | 15 (5%) / 33 (11%) | 15 (6%) / 30 (11%) | 0 (0%) /3 (13%) | .45 |
| | KRAS | 15 (5%) / 14 (5%) | 13 (5%) / 13 (5%) | 2 (8%)/ 1 (4%) | |
| | ROS1 | 4 (1%) / 12 (4%) | 3 (1%) / 11 (4%) | 1(4%) / 1 (4%) | |
| Comorbidities | Diabetes | 56 (19%) | 44 (16%) | 12 (50%) | .0002 |
| | HIV | 9 (3%) | 9 (3%) | 0 (0%) | .77 |
| | Heart disease | 60 (20%) | 53 (20%) | 7 (29%) | .39 |
| | Kidney disease | 52 (18%) | 45 (17%) | 7 (29%) | .21 |
| | Liver disease | 9 (31%) | 8 (3%) | 1 (4%) | 1 |
| 1st line treatment approach | Surgery only | 69 (23%) | 63 (23%) | 6 (25%) | .85 |
| | Surgery + adjuvant / neoadjuvant platinum based CTX | 101 (34%) | 93 (34%) | 8 (33%) | .85 |
| | RCTX | 11 (4%) | 11 (4%) | 0 (0%) | .31 |
| | Target Therapy | 4 (1%) | 3 (1%) | 1 (4%) | .21 |
| | Platinum-based CTX | 95 (32%) | 87 (32%) | 8 (33%) | .68 |
| | Other | 5 (2%) | 5 (2%) | 0 (0%) | .50 |
| | BSC | 5 (2%) | 5 (2%) | 0 (0%) | .50 |
| | Unknown | 5 (2%) | 4 (1%) | 1 (0%) | .35 |

Count data is shown unless indicated otherwise. *Differences between colonized and non-colonized patients were tested. Mann-Whitney U test was used to calculate P value for age. Except for EGFR, gene mutations were not tested due to missing data. CT, chemotherapy; TKI, tyrosine kinase inhibitor; NSCLC, non- small cell lung cancer; SCNSCLC, squamous cell NSCLC.

colonized with MDRO (MDRO^pos). Two hundred seventy-one patients (91.9%) were defined as MDRO^neg within the screening period. Median time to first MDRO screening calculated from first diagnosis was 20 days (range, 0–84 days). Comparative descriptive statistics of the study groups are illustrated in Table 1. Median age was 66 years (range, 29–90 years). Approximately 80% had an ECOG performance status of 0 or 1 and one third of all patients presented with metastatic disease stage (UICC IV). The majority of patients were former or active

smokers. Aside of concomitantly underlying diabetes mellitus that was more frequently present in MDRO[pos] patients, we did not find significant differences in patient or disease characteristics between MDRO[pos] and MDRO[neg] patients in univariate and multivariate analysis (S2 Table in S1 File).

First-line treatment approaches did not differ significantly between study groups. Notably, only a minority of patients diagnosed with driver mutations received a first-line targeted therapy. This is partially owed to the fact that ALK, ROS1 and BRAF inhibitors were first approved for first line treatment in Germany in late 2016 and 2018, respectively. Five patients in the MDRO[neg] group and no patient in the MDRO[pos] group received best supportive care only.

We then compared the eligible study cohort with patients identified in the registry without MDRO screening within the predefined time frame. The off-target population (107/402; 26.6%) was significantly younger (p = 0.001), had a higher proportion of patients with ECOG 3 or worse performance status in addition to a higher proportion of patients with advanced or metastatic disease (p = 0.0001) (S3 Table in S1 File). Besides diabetes, which was more prevalent in the study cohort (p = 0.004), other comorbidities were well balanced. The OS of the off-target cohort was significantly inferior compared to the study cohort, yet no survival differences in patients with advanced or metastatic disease (IIIB, IV; UICC 7[th]) between the overall off-target and the study population were noticed (not shown).

## MDRO

A total of 24 patients (8.1%) were screened positive for MDRO colonization. Detailed information on resistance phenotype of all MDRO is shown in S4 Table in S1 File. *Enterobacterales* were by far the most frequent MDRO detected with a proportion of 79.2% (19/24), all of which had phenotypical resistance to 3[rd]/4[th] generation Cephalosporins (Ceftriaxone, Cefotaxime, Ceftazidime, Cefepime). Additionally, most species were resistant to piperacillin and more than half were resistant to folate pathway inhibitors (Trimethoprim/Sulfamethoxazole). Resistance against aminoglycosides (Amikacin, Gentamicin), tigecycline and fosfomycin were infrequent. All MDR *Enterobacterales* detected were susceptible to carbapenems (Imipenem, Meropenem, Ertapemem). *Enterococcus faecium* with resistance to ampicillin, carbapenem and fluoroquinolones (Levofloxacine, Ciprofloxacine, Moxifloxacine) and incomplete resistance to glycopeptides (Vancomycine, Teicoplanin)(3x vanB phenotype, 1x vanA phenotype) were detected in 16.7% (4/24) of all MDRO[pos] cases. Additional resistance to aminoglycosides (high-level) and tetracyclines was detected in one case each. One MRSA (4.2%, 1/24) with phenotypical resistance against fluoroquinolones, lincosamides (Clindamycin) and macrolides (Erythromycin) was identified. The most common location for MDRO colonization was rectal (95.8%) in all but the MRSA case, which was detected in a nose swab.

The incidence of subsequent colonization with multiple MDRO in MDRO[pos] patients within the screening period was 25%, 3 patients acquired additional ESBL-producing species and 3 patients acquired additional VRE. Altogether, 16 patients in the MDRO[neg] group were subsequently screened positive for MDRO after a median time calculated from first diagnosis of 495 days (range, 109–1231 days). Because subsequent screening procedures in patients with NSCLC were only irregularly performed, especially in patients who were mainly treated on an outpatient basis, further analyses on these patients (with subsequently acquired MDRO colonization) were not carried out due to probable selection bias of this subpopulation.

## Primary outcome analysis: Survival

Kaplan-Meier estimates for EFS and OS of the overall population and stratified by MDRO colonization are shown in Fig 1A–1D. Median EFS did not differ between MDRO[pos] (7.1 months;

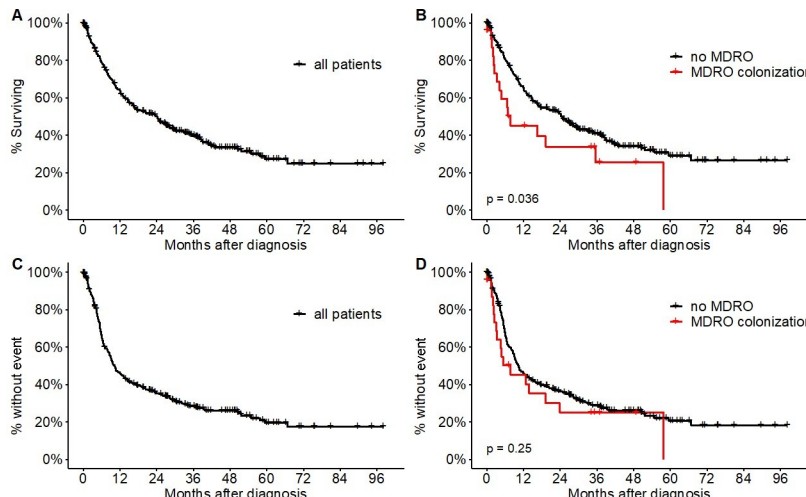

**Fig 1. Kaplan-Meier estimates of overall survival (OS) and progression-free survival (PFS).** (A) OS of all patients. (B) OS of patients stratified by multidrug-resistant organism (MDRO) colonization. (C) PFS of all patients. (D) PFS of patients stratified by colonization with MDRO. Log-rank test was used to calculate p values in (C+D).

95% CI, 0.0–16.7 months) and MDRO$^{neg}$ (10.3 months; 95% CI, 7.9–12.9 months) study groups with a hazard ratio (HR) of 1.25 (95% CI, 0.74–2.21; p = 0.25) (Fig 1D), that was further confirmed by multivariate analysis (S5 Table in S1 File). There were 92 censored events (31.2%) in the EFS analysis. Median OS was significantly inferior in the MDRO$^{pos}$ study group with a median OS of 7.8 months (95% CI, 0.0–19.9 months) compared to a median OS of 23.9 months (95% CI, 17.6–30.1 months) in the MDRO$^{neg}$ group resulting in a HR of 1.9 (95% CI, 1.02–3.7); p = 0.036)(Fig 1B). There were 120 censored events (41.0%) in the OS analysis.

When stratified for disease stage (Fig 2A), median OS in the MDRO$^{pos}$ study group showed a significantly inferior median survival time in patients with advanced (IIIB) or metastatic

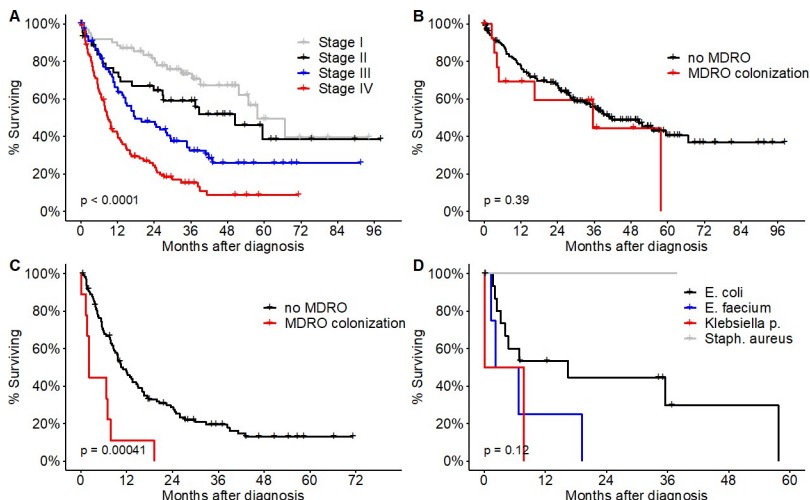

**Fig 2. Competing risks analysis for death.** (A) Cumulative incidence functions for relapse mortality, non-relapse mortality, or mortality not otherwise specified (unknown) of all patients. (B) Cumulative incidence functions of patients stratified by multidrug-resistant organism (MDRO) colonization. Competing risks regression model [35] was used to calculate p values for differences in non-relapse mortality (p < 0.00001) and relapse mortality (p = 0.49) between patients colonized by MDRO and patients without MDRO.

| Risk Factor | Comparator | | HR | 95% CI | P value |
|---|---|---|---|---|---|
| MDRO colonization | No colonization | | 1.96 | (1.09—3.51) | .02 |
| Gender: male | Gender: female | | 1.22 | (0.86—1.71) | .26 |
| Age at diagnosis | | | 1.0 | (0.99—1.01) | .16 |
| Extensive disease | Limited disease | | 3.03 | (2.16—4.24) | <.001 |
| Histology: SCC | Histology: adenocarcinoma | | 1.23 | (0.88—1.72) | .23 |
| Histology: others | Histology: adenocarcinoma | | 0.63 | (0.20—2.00) | .43 |
| ECOG: 1 | ECOG: 0 | | 1.40 | (0.93—2.09) | .11 |
| ECOG: 2 | ECOG: 0 | | 2.44 | (1.52—3.93) | <.001 |
| ECOG: 3 | ECOG: 0 | | 6.63 | (1.98—22.27) | .002 |
| ECOG: 4 | ECOG: 0 | | 1.08 | (0.23—5.16) | .92 |
| Diabetes mellitus | No diabetes mellitus | | 1.10 | (0.74—1.64) | .64 |

Cox proportional hazard regression analysis. MDRO, multidrug resistant organisms; SCC, squamous cell carcinoma; HR, hazard ratio; CI, confidence interval.

**Fig 3. Multivariate analysis of risk factors for death.**

disease (IV)(4.4 months vs 10.5 months; HR, 2.9; 95%CI, 1.9–19.6; p = 0.0004) (Fig 2C), whereas we found no significant difference in survival between MDRO[pos] and MDRO[neg] study groups with early disease stages (IA-IIIA; HR 1.4; 95%CI, 0.6–3.5; p = 0.39)(Fig 2B). Stratification by MDRO species did not yield significant differences in OS among MDRO[pos] patients colonized with VRE, MRSA or ESBL (p = 0.12) (Fig 2D). The negative impact on survival outcomes was further confirmed in multivariate analysis adjusted for gender, age, disease stage, ECOG performance status, NSCLC histology and presence of concomitant underlying diabetes (Fig 3). In addition to MDRO colonization, performance status and disease stage were identified as independent prognostic variables.

## Secondary and exploratory outcome analysis

**Cause of death.** The distribution of causes of death stratified by MDRO colonization status is depicted in Table 2 and Fig 4. There was a significantly higher rate of non-cancer-related-mortality in MDRO[pos] patients compared to MDRO[neg] patients (p = 0.002) and a significantly higher rate of infectious causes (p = 0.002) The most frequently observed infection-related cause of death was pneumonia with or without septicemia in 5 cases in the MDRO[pos] group, 2 additional patients died of pleural empyema. The empirical antibiotic treatment approach in 5 of these patients consisted of agents that were primarily tested non-susceptible to the detected MDRO. Invasive infections from the formerly detected MDRO within the

**Table 2. Equal distribution of causes of death between the MRDO[pos] and MRDO[neg] subgroup.**

| | | no MDRO (n = 271) | MDRO colonization (n = 24) | P value |
|---|---|---|---|---|
| NCR | | 22 (8%) | 9 (38%) | .0002 |
| | infectious related | 8 (36%) | 7 (78%) | .0002 |
| | cardiovascular disease related | 9 (41%) | 2 (22%) | .24 |
| | major bleeding with fatal outcome | 4 (18% | 0 (0%) | 1.0 |
| | Asphyxia | 1 (5%) | 0 (0%) | 1.0 |
| CR | | 107 (39%) | 6 (25%) | .40 |
| UKN | | 29 (10%) | 1 (4%) | .49 |
| LTFU | | 113 (41%) | 8 (33%) | |

Causes of death were compared using fishers exact test. NCR, non-cancer related mortality; CR, cancer-related; UKN, unknown; LTFU, lost to follow up; MDRO Multi Drug Resistant Organism; assessed using Fisher's Exact Test.

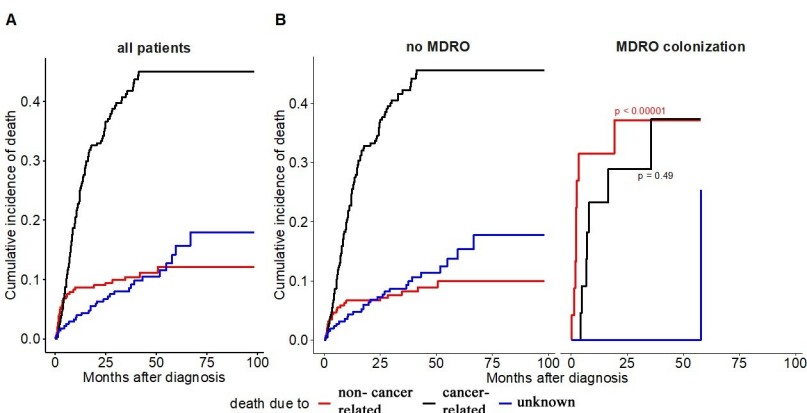

**Fig 4.** Cumulative incidence of death stratified by non-cancer related and cancer related mortality (A) in the whole study group. (B) stratified by MDRO colonization.

MDRO[pos] group were determined in two cases (2/7, 28.6%) (VRE-positive blood culture of a patient with pneumonia-induced sepsis; evidence of ESBL in pleural empyema). In the remaining 5 patients the pathogenic organism could not be detected by serial blood cultures.

In the MDRO[neg] study group, 8 patients (36%) succumbed to infectious complications, 4 of which had evidence of an invasive pathogen. One of these patients died of pneumonia-induced sepsis caused by a subsequently acquired (after the initial screening period) piperacillin- and carbapenem-resistant *Pseudomonas aeruginosa*, whose profile of resistance could not be considered at the time of initial antibiotic treatment.

**Number and duration of hospital stays.** Overall, there were no differences in number and duration of all-cause hospital admissions between MDRO[pos] and MDRO[neg] patients. Likewise, there were no differences in number and duration of hospital admissions for infectious complications between MDRO[pos] and MDRO[neg] study groups (S6 and S7 Figs in S1 File). Comparison of number and duration of inpatient treatments between study groups were however not adjusted for differences in median survival times between MDRO[pos] and MDRO[neg] patients.

## Discussion

To our knowledge, this is the first study that aimed to determine the clinical impact of MDRO colonization in patients with NSCLC. We show that MDRO colonization is an independent risk factor for impaired overall survival, independent of confounding variables, such as performance status and disease stage.

Our study demonstrates considerable colonization rates (8.1%) with ESBL producing *Enterobacterales* and VRE species in patients with NSCLC across all subgroups in terms of age, stage, performance status and concomitant underlying (renal, heart, liver) diseases among other variables. We encountered a significantly higher co-occurrence of diabetes in patients screened positive for MDRO. Diabetes has previously been identified as a potential risk factor for MDRO colonization [36, 37] and subsequent bloodstream infections with intestinal bacteria due to disruption of the gut barrier [38, 39]. The overall prevalence of MDRO colonization at admission has been reported to be as high as 10% for ESBL producing *Enterobacterales* [40, 41], reaching a prevalence of 20% in specific patient subgroups [9], and 2% for VRE [42] in German tertiary care centers. The colonization rate in our study was slightly lower than previously reported. Colonization rates are known to be significantly influenced by the patient subgroups examined and other risk factors such as antibiotic and surgical pretreatment, proton

pump inhibitor usage, travel habits, prior hospitalizations and country of origin [33, 40, 43–45]. These factors were not assessed in our study and might contribute to the lower prevalence of MDRO colonization seen in our cohort. Furthermore, as many patients are seen as outpatients (with less stringent screening), MDRO positive patients may be underreported.

Approximately 80% of non-cancer-related mortality in the MDRO[pos] group was infection-related as extracted from the corresponding death certificates. We did not observe any differences in hospital admission rates and/or duration of inpatient treatment (for infectious or other causes) between MDRO[pos] and MDRO[neg] patients, suggesting that MDRO colonization by itself may not be a strong risk factor for the frequency of subsequent invasive bacterial infections in this patient cohort, but instead mediates a higher fatality rate due to more severe infectious complications. However, this data is hard to interpret. Firstly, the number of outpatient visits (e.g. for infectious complications) could not be analyzed due to insufficient documentation. Secondly, we *do not have sufficient information on the final course of each individual patient to judge the contribution of infectious-related complications to the death of patients with progressive cancer. And thirdly, we cannot exclude a misclassification of the cause of death by the responsible physician.*

Infections, particularly involving the lung tissue have been identified as a major cause of death in several retrospective studies [28, 29]. Patients with advanced disease stages were more prone to infectious complication and data suggests that they may adversely affect survival.

It has been shown that the increased fatality rate in MDRO[pos] patients is at least partially attributable to inadequate empirical antibiotic treatment in case of invasive infections [17, 46]. Indeed, in 5 of the 7 fatal infections within the MDRO[pos] cohort, the initial antibiotic regime did not take into account the prior proven MDRO colonization. Colonizing MDR bacteria were detected in 2 out of the 7 cases (29%) of pulmonary infections reported here. This is in agreement with previous reports on the overall low sensitivity regarding the detection of invasive pathogens by blood cultures [47]. Bacteremia is diagnosed in less than 10% by serial blood cultures of patients suffering from pneumonia despite clinical indications of bloodstream infections. Nevertheless, gut bacteria play a major role in NSCLC-associated lung tissue infections [48–50] and empirical antibiotic treatment should be selected considering intestinal MDRO bacteria.

There is emerging evidence that the gut microbiota affects systemic inflammation and immunity and there are multiple possible mechanisms linking microbiota to carcinogenesis, tumor outgrowth and metastases, altered metabolism, pro-inflammatory and impaired immune-response [51–53]. Almost all colonizing MDRO in our study have been identified by rectal screening. Susceptibility to and presence of intestinal MDRO has been linked to alterations in the gut microbiota with reduced bacterial diversity [54, 55], which in turn is associated with reduced tumor response to cytotoxic agents and immunotherapy in lung cancer [56–59]. This is also supported by reduced clinical benefit from immunotherapy after the usage of antibiotics in patients with NSCLC [56, 60]. In our study, however, first-line EFS was not different between MDRO[pos] and MDRO[neg] groups, indicating only minor–if any–influence of MDRO on response to conventional antineoplastic therapy. As immunotherapeutic agents were not approved for first-line treatment in NSCLC until 2017, we cannot draw conclusions regarding the impact of MDRO colonization on the treatment response to immunotherapeutic agents. Prospective studies are needed to further address the relevance of MDRO colonization and the impact of intestinal microbiota alterations on tumor response to immunotherapy and/or cytotoxic agents.

Finally, there is conflicting evidence, whether MDR bacteria have additional genomic content including factors known or supposed to be associated with increased virulence [61, 62]. Vancomyin-resistant *E. faecium* and ESBL-producing species have been shown to incorporate

virulence factors in co-occurrence with genes for antibiotic resistance [63–68] and these factors might overall contribute to the higher mortality seen in MDRO[pos] patients. However, since genetic analyses addressing the co-occurrence of virulence factors other than antibiotic resistance genes were not performed, we can only speculate on their influence on the overall mortality outcome in our study.

We fully acknowledge the limitations of a retrospective analysis conducted in a single tertiary treatment center. Significant differences between the study group and off-target population are indicative of selection bias due to MDRO screening. However, the proportion of patients excluded from the final analysis due to missing MDRO screening was only approximately 25% of the total screening population (patients with second malignancy excluded). Additionally, due the overall limited sample size our results need confirmation in larger series before drawing final conclusions regarding the impact of MDRO colonization in patients with oncological diseases. However, we believe that our findings corroborate available data collected in patients with (dominantly) hematologic malignancies that consistently show inferior survival outcomes in patients either with invasive MDRO infections or MDRO colonization [3, 5, 9, 12–14, 17, 69–71].

## Conclusion

We conclude that MDRO colonization our population is an independent risk factor for inferior OS in patients diagnosed with NSCLC. Impairment Patients with advanced or metastatic disease seem to be at highest risk for impaired survival. Furthermore our data suggest, that a higher rate of non-cancer related mortality and infections in particular might contribute to the inferior survival in MDRO colonized patients. Given the high and rising rate of MDRO colonization in oncological patients, early and frequent screening is warranted in both outpatient and inpatient settings. Empirical antibiotic treatment approaches need to cover formerly detected MDR commensals in cases of (suspected) invasive infections.

More studies should elucidate the impact of MDRO colonization and intestinal bacterial diversity within the rapidly changing landscape of antineoplastic treatment options in patients with NSCLC.

## Supporting information

**S1 Table.**
(XLSX)

**S1 File.**
(DOCX)

## Acknowledgments

We thank all members of and contributors to the University Cancer Center Frankfurt, who made this analysis possible.

## Author Contributions

**Conceptualization:** Jan A. Stratmann, Olivier Ballo, Shabnam Shaid, Sebastian Koschade.

**Data curation:** Jan A. Stratmann, Raphael Lacko, Olivier Ballo, Shabnam Shaid, Wolfgang Gleiber, Peter Kleine, Susanne Stera, Sebastian Koschade.

**Formal analysis:** Jan A. Stratmann, Raphael Lacko, Maria J. G. T. Vehreschild, Christian Brandts, Sebastian Koschade.

**Investigation:** Jan A. Stratmann, Wolfgang Gleiber, Peter Kleine, Susanne Stera.

**Methodology:** Jan A. Stratmann, Shabnam Shaid, Maria J. G. T. Vehreschild, Thomas Wichelhaus, Claudia Reinheimer, Stephan Göttig, Volkhard A. J. Kempf.

**Project administration:** Jan A. Stratmann, Christian Brandts.

**Software:** Sebastian Koschade.

**Supervision:** Maria J. G. T. Vehreschild, Thomas Wichelhaus, Claudia Reinheimer, Stephan Göttig, Volkhard A. J. Kempf, Christian Brandts, Martin Sebastian, Sebastian Koschade.

**Validation:** Thomas Wichelhaus, Martin Sebastian.

**Writing – original draft:** Jan A. Stratmann.

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
