## [Decision Letter · Decision Letter 0]

10 Aug 2020

PONE-D-20-09321

Colonization with multi-drug-resistant organisms negatively impacts survival in patients with non-small cell lung cancer

PLOS ONE

Dear Dr. Stratmann,

Thank you for submitting your manuscript to PLOS ONE. After careful consideration, we feel that it has merit but does not fully meet PLOS ONE’s publication criteria as it currently stands. Therefore, we invite you to submit a revised version of the manuscript that addresses the points raised during the review process.

Specifically, the reviewers have raised overlapping concerns about the reporting of the Results and statistical methodology as well as the conclusions presented in the Discussion section.

We look forward to receiving your revised manuscript.

Kind regards,

Richard Hodge

Associate Editor

PLOS ONE

Journal Requirements:

2. Thank you for including your competing interests statement; "I have read the journal's policy and the authors of this manuscript have the following competing interests: JS reports personal fees from Bristol-Myers & Squibb, personal fees from Novartis, personal fees from Roche, outside the submitted work. RL has nothing to disclose. OB has nothing to disclose. SS has nothing to disclose. WG has nothing to disclose. MJGTV has served at the speakers’ bureau of Akademie für Infektionsmedizin, Ärztekammer Nordrhein, Astellas Pharma, Basilea, Gilead Sciences, Merck/MSD, Organobalance, Pfizer and Uniklinik Freiburg / Kongress und Kommunikation, received research funding from 3M, Astellas Pharma, DaVolterra, Gilead Sciences, MaaT Pharma, Merck/MSD, Morphochem, Organobalance, Seres Therapeutics, and is a consultant to Alb-Fils Kliniken GmbH, Ardeypharm, Astellas Pharma, Berlin Chemie, DaVolterra, Ferring, MaaT Pharma and Merck/MSD. TW has nothing to disclose. CR has nothing to disclose. SG has nothing to disclose. VK reports grants and personal fees from DFG Germany (DFG FG 2251), grants and personal fees from EU Marie Sklodowska-Curie (#765042), during the conduct of the study. PK has nothing to disclose. SS has nothing to discose. CB has nothing to disclose. MSe reports personal fees from Lilly, personal fees from Astra-Zeneca, personal fees from Bristol-Myers & Squibb, personal fees from Merck Sharp & Dohme, personal fees from Pfizer, personal fees from Takeda, personal fees from Roche, personal fees from AbbVie, personal fees from Boehringer-Ingelheim, personal fees from Celgene, personal fees from Novartis, outside the submitted work. SK has nothing to disclose."

Reviewers' comments:

Reviewer's Responses to Questions

**Comments to the Author**

1. Is the manuscript technically sound, and do the data support the conclusions?

Reviewer #1: Partly

Reviewer #2: Yes

Reviewer #3: No

Reviewer #4: Yes

2. Has the statistical analysis been performed appropriately and rigorously? 

Reviewer #1: Yes

Reviewer #2: Yes

Reviewer #3: N/A

Reviewer #4: Yes

3. Have the authors made all data underlying the findings in their manuscript fully available?

Reviewer #1: Yes

Reviewer #2: Yes

Reviewer #3: No

Reviewer #4: No

4. Is the manuscript presented in an intelligible fashion and written in standard English?

Reviewer #1: No

Reviewer #2: Yes

Reviewer #3: No

Reviewer #4: Yes

5. Review Comments to the Author

Reviewer #1: The manuscript titled 'Colonization with multi-drug-resistant organisms negatively impacts survival in patients with non-small cell lung cancer' tries to show the importance of multi-drug resistant organisms on survival and mortality of patients suffering from non-small cell lung cancer. The main limitation of this investigation is the use of a very small sample to draw an important/significant conclusion.

Major comments:

1. The study sample size is small, only 24 out of 295 (8%) patients were studied. The sample size must be significantly larger than this to draw any valid conclusion.

2. In Pg 2 Lines 42-46, the authors mention that 'There was a significantly higher rate of non-cancer-related-mortality in MDROpos patients compared to MDROneg patients (p<0.001) with a trend towards an increased rate of fatal infections in MDROpos patients (p=0.05).

Also, Pg 11, Lines 231-232 states 'When stratified for disease stage (Fig. 2A), median OS in the MDROpos study group showed a significantly inferior median survival time in patients with advanced (IIIB) or metastatic disease (IV). Then again, in Pg 14, Line 302, the authors say ‘Approximately 80% of non-relapse mortality in the MDROpos group was infection-related.’

These are strong statements. Patients in advanced stages of cancer are more likely to suffer from mortality when compared to those in early stages of cancer simply because of their physiological conditions and not just because of bacterial colonization.

Bacterial infections in patients at early stage of cancer can actually augment progression to more critical stages/metastasis according to the existing literature. So, it is the severity of the infection at the early stage which determines or is one of the predisposing factors to progression to a more critical stage.

3. There are many existing literature that the authors can compare their work with such as:

i. Sarihan S, Ercan I, Saran A, Cetintas SK, Akalin H, Engin K. Evaluation of infections in non-small cell lung cancer patients treated with radiotherapy. Cancer Detect Prev. 2005;29(2):181-188. doi:10.1016/j.cdp.2004.11.001

ii. https://www.cancernetwork.com/view/infectious-complications-lung-cancer.

4. Recent literature suggests the presence of the bacterial genus Acidovorax particularly in lung cancer patients. In the manuscript, the authors did not provide details of the bacterial genera found. The author might consider comparing their findings with the paper and comment on why their microbiome could be different:

Greathouse, K.L., White, J.R., Vargas, A.J. et al. Interaction between the microbiome and TP53 in human lung cancer. Genome Biol 19, 123 (2018). https://doi.org/10.1186/s13059-018-1501-6.

5. Since the authors emphasize on the effect of bacteria on mortality, the author must include one paragraph on what has already been published on the possible role of microbial infections on mortality of cancer patients. Recent literature such as the following could have been cited.

Translational Oncology. VOLUME 14, ISSUE 12, P2097-2108, DECEMBER 01, 2019

Gram-Negative Pneumonia Augments Non–Small Cell Lung Cancer Metastasis through Host Toll-like Receptor 4. Stephen D. Gowing, Simon C. Chow, Jonathan J. Cools-Lartigue, Simon Rousseau, Salman T. Qureshi, Lorenzo E. Ferri, https://doi.org/10.1016/j.jtho.2019.07.023

Koslow M, Epstein Shochet G, Matveychuk A, Israeli-Shani L, Guber A, Shitrit D. The role of bacterial culture by bronchoscopy in patients with lung cancer: a prospective study. J Thorac Dis. 2017;9(12):5300-5305. doi:10.21037/jtd.2017.10.150

Ye M, Gu X, Han Y, Jin M, Ren T. Gram-negative bacteria facilitate tumor outgrowth and metastasis by promoting lipid synthesis in lung cancer patients. J Thorac Dis. 2016;8(8):1943-1955. doi:10.21037/jtd.2016.06.47

Kovaleva, O.V.; Romashin, D.; Zborovskaya, I.B.; Davydov, M.M.; Shogenov, M.S.; Gratchev, A. Human lung microbiome on the way to cancer. J. Immunol. Res. 2019, 2019, 1394191.

Chow, S. C. et al. Gram negative bacteria increase non-small cell lung cancer metastasis via Toll-like receptor 4 activation and mitogen-activated protein kinase phosphorylation. Int J Cancer 136, 1341–1350, https://doi.org/10.1002/ijc.29111 (2015).

Minor corrections:

6. In Pg 3 line 56 cephalosporine spelling correction to Cephalosporin

7. In Pg 4 Line 83 says '(definition see below)' – no definition given

8. In Pg 4 Line 84 ‘…were included into this analysis…’ consider changing the word 'into' to 'to'.

9. In Pg 4 Line 85 ‘….second malignancy - aside of localized non-melanoma skin cancer….’ consider replacing ‘aside of’ by ‘aside from’

10. In Pg 5 Line 100 ‘…….unit as well as all patients all patients admitted to the thoracic surgery…..’, repetition of ‘all

patients’.

11. In Pg 14, Lines 289-290, ‘We encountered a significantly higher co-occurence of diabetes in patients’ spelling of occurrence.

Reviewer #2: I would like to congratulate the authors for conducting the study. It is a well written protocol. There are no previous studies on MDRO and lung cancer. The authors have clearly discussed the limitations.

Reviewer #3: The authors aimed to reveal that the impact of MDRO colonization in patients who have been diagnosed with Non-small cell lung cancer (NSCLC) who are at known high-risk for invasive infections. However, the data is not accurate enough. The data in this manuscript do not support the conclusion.

Reviewer #4: Authors have presented a very important topic on antimicrobial resistance in patients with NSCLC. Antibiotic resistance is a global problem which need be to tackled worldwide. Despite low prevalence of MDRO in their study, however this data from developed world merit to be shared to scientific community at large.

Overall the manuscript is well written and interesting to read. However, results and discussion sections need to revised. In results section some tables are difficult to understand and statistical interpretations are not well understood.

Specific comment

Abstract:

Line 38: Replace 295 with Two hundred and ninety-five.

Line 44: Add crude and adjusted, plus their 95%CI for both univariate and multivariable analysis, respectively.

Line 44 – 46: “There was a significantly higher rate of non-cancer-related mortality in MDROpos compared to MDROneg patients (p<0.001) with a trend MDROpos towards an increased rate of fatal infections in patients (p=0.05)” from table 3 I found very difficult to interpret. The way p-value which has been presented it looks to reject null hypothesis in favor of alternative hypothesis. Considering NRM cases alone in comparison of non-cancer related mortality might obscure the intended outcome, if possible include all participants in this analysis. Table 3 need to be revised for more clarity.

Introduction:

Line 55–56: Third-generation cephalosporin resistant should be replaced with extended spectrum beta lactamase producers.

Line 57: Replace Staphylococcus aureus with methicillin resistance to Methicillin resistance Staphylococcus aureus

Materials and Methods

ESBL, MRSA and VRE were screened by screening media, these media have high sensitivity. It will be interesting to state how this MDRO were confirmed either by phenotypic or genotypic.

Line 146: Ensure consistency either use MDROpos or MDRO+ .

Results

Line 168: Replace 271 with Two hundred and seventy-one.

Line 173: Diabetes mellitus did not show significant association, revisit supplementary file OR included 1.

Table1: Some variable in a column total % do not add to 100%. For example, in co-morbidity variable total number of MDRO is 25 and not 24. Interpretation along the row could be interesting than along the column. Think of revising this table.

Line 197 – 207: Specify name of specific antibiotics tested instead of using classes of antibiotics. For examples macrolide use either erythromycin or azithromycin, aminoglycosides specify gentamicin or others.

In analysis MDRO+ exposure to OS outcome. The disease stage could be one of the important confounders for OS. See Fig1A for example, for interest I would like to know if you controlled for disease stage as confounder, what happen HR. If you did not do please could you explain to me for interest.

Although this has been explained in discussion, however authors need to dig further on MDRO colonization with inferior overall survival outcome. It is well hypothesized MDRO colonization is a risk for severe MDRO infection with the same bacteria. MDRO infection could have direct effect on overall survival in these group of patients.

Analysis on Fig2B and 2C could be combined (IV/IIIB vs IIIA/II/I) rather than categorized in different group. Since disease staging in itself could predict OS.

Line 246 – 251 looks like figure 2 legend, please move below the figure2

Line 256 – 258: Check comment on abstract.

Table 3: Check comment on abstract

Line 267 – 268: Looks like legend for figure 3.

Discussion:

Well written, however in some part authors need to revise like line 302. If table 3 is revised this statement might need to change, there is a need to include all mortality not NRM only.

In conclusion, line 350 authors did not establish the correlation between MDRO colonization and infection. To state this as a reason “due higher rate of fatal infections mostly involving the lung tissue”. This infection was unrelated to colonization, the statement needs to be revised.

6. PLOS authors have the option to publish the peer review history of their article (what does this mean?). If published, this will include your full peer review and any attached files.

Reviewer #1: **Yes: **Professor Sunjukta Ahsan, Department of Microbiology, University of Dhaka, Bangladesh, sunjukta@du.ac.bd

Reviewer #2: No

Reviewer #3: No

Reviewer #4: No

---

## [Author Response · Author response to Decision Letter 0]

13 Oct 2020

Dear Prof. Heber, Dear Richard Hodge

thank you for the possibility to re-submit our manuscript after major revision. We thank the referees for their valuable comments and have addressed all their concerns. 

We hope the manuscript can now be accepted in Plos One and look forward to your response. 

Sincerely,

Jan Stratmann, Sebastian Koschade

Comments to the referees:

Reviewer #1: The manuscript titled 'Colonization with multi-drug-resistant organisms negatively impacts survival in patients with non-small cell lung cancer' tries to show the importance of multi-drug resistant organisms on survival and mortality of patients suffering from non-small cell lung cancer. The main limitation of this investigation is the use of a very small sample to draw an important/significant conclusion.

Major comments:

1. The study sample size is small, only 24 out of 295 (8%) patients were studied. The sample size must be significantly larger than this to draw any valid conclusion.

We agree with the reviewer that the overall sample size is limited. Albeit conclusions derived from our work need confirmation in larger studies, we still believe our data is of importance for our readership, particularly due to increasing prevalence of MDRO worldwide and secondly due to the wide lack of clinical evidence regarding the impact of MDROs in oncological diseases. 

We now acknowledge the limited sample size in the discussion section.

2. In Pg 2 Lines 42-46, the authors mention that 'There was a significantly higher rate of non-cancer-related-mortality in MDROpos patients compared to MDROneg patients (p<0.001) with a trend towards an increased rate of fatal infections in MDROpos patients (p=0.05). Also, Pg 11, Lines 231-232 states 'When stratified for disease stage (Fig. 2A), median OS in the MDROpos study group showed a significantly inferior median survival time in patients with advanced (IIIB) or metastatic disease (IV). Then again, in Pg 14, Line 302, the authors say ‘Approximately 80% of non-relapse mortality in the MDROpos group was infection-related.’

These are strong statements. Patients in advanced stages of cancer are more likely to suffer from mortality when compared to those in early stages of cancer simply because of their physiological conditions and not just because of bacterial colonization.

Bacterial infections in patients at early stage of cancer can actually augment progression to more critical stages/metastasis according to the existing literature. So, it is the severity of the infection at the early stage which determines or is one of the predisposing factors to progression to a more critical stage.

We agree with the reviewer and have revised the relevant sections. While suggestive, our retrospective trial design does not allow us to identify causal factors. In order to address this justified criticism, we mitigated our claims suggestive of causality throughout the manuscript. However, this does not pertain to our main finding of significantly poorer overall survival in MDROpos NSCLC patients (primary study outcome). 

We agree with the notion that infection-related complications might also be a clinically relevant factor in relapsed/progressed patients and that an analysis of the distribution of infections between these patients might give additional information. However, we relied on the cause of death as described in the final medical report or on the death certificate. We unfortunately do not have sufficient information on the final course of each individual patient to judge the contribution of infectious-related complications to the death of patients with progressive cancer. However, we believe that information gained from this secondary/exploratory analysis is rather hypothesis generating than convincing evidence, we therefore – as mentioned above - weakened our claims regarding this analysis throughout the manuscript.

Regarding the impact on survival we have already adjusted for disease stage in our multivariate Cox proportional hazards regression analysis of overall survival (see Tab. 2, limited vs extensive disease as well as Fig. 2). As expected, extensive disease was highly significantly associated with a poorer OS (HR 3.03, 95% CI 2.16–4.24). The association between MDRO colonization and OS still remained significant in this multivariable analysis in the overall cohort (Tab. 2).

3. There are many existing literature that the authors can compare their work with such as:

i. Sarihan S, Ercan I, Saran A, Cetintas SK, Akalin H, Engin K. Evaluation of infections in non-small cell lung cancer patients treated with radiotherapy. Cancer Detect Prev. 2005;29(2):181-188. doi:10.1016/j.cdp.2004.11.001

ii. https://www.cancernetwork.com/view/infectious-complications-lung-cancer.

As we have pointed out in the introduction, infections are important complications in patients with lung cancer. We thank the reviewer for his suggestions on additional literature regarding (pulmonary) infections in these patients and have added more background data mainly to the introduction section.

4. Recent literature suggests the presence of the bacterial genus Acidovorax particularly in lung cancer patients. In the manuscript, the authors did not provide details of the bacterial genera found. The author might consider comparing their findings with the paper and comment on why their microbiome could be different:

Greathouse, K.L., White, J.R., Vargas, A.J. et al. Interaction between the microbiome and TP53 in human lung cancer. Genome Biol 19, 123 (2018). https://doi.org/10.1186/s13059-018-1501-6.

We highly appreciate the reviewer´s suggestion to compare the above-mentioned literature which is focusing on microbial characteristics / the bacterial consortium within the lung cancer (intratumoral) microenvironment. Contrary to Greathouse et al, we focused on the colonization with MDRO species that are detected during clinical routine. The microbial diversity either in the lung or gut in patients with specific cancer types is a competitive field of research at present, it is however not part of our research and data. Detailed data regarding the MDRO species in our study can be found in Table S4.

5. Since the authors emphasize on the effect of bacteria on mortality, the author must include one paragraph on what has already been published on the possible role of microbial infections on mortality of cancer patients. Recent literature such as the following could have been cited.

Translational Oncology. VOLUME 14, ISSUE 12, P2097-2108, DECEMBER 01, 2019

Gram-Negative Pneumonia Augments Non–Small Cell Lung Cancer Metastasis through Host Toll-like Receptor 4. Stephen D. Gowing, Simon C. Chow, Jonathan J. Cools-Lartigue, Simon Rousseau, Salman T. Qureshi, Lorenzo E. Ferri, https://doi.org/10.1016/j.jtho.2019.07.023

Koslow M, Epstein Shochet G, Matveychuk A, Israeli-Shani L, Guber A, Shitrit D. The role of bacterial culture by bronchoscopy in patients with lung cancer: a prospective study. J Thorac Dis. 2017;9(12):5300-5305. doi:10.21037/jtd.2017.10.150

Ye M, Gu X, Han Y, Jin M, Ren T. Gram-negative bacteria facilitate tumor outgrowth and metastasis by promoting lipid synthesis in lung cancer patients. J Thorac Dis. 2016;8(8):1943-1955. doi:10.21037/jtd.2016.06.47

Kovaleva, O.V.; Romashin, D.; Zborovskaya, I.B.; Davydov, M.M.; Shogenov, M.S.; Gratchev, A. Human lung microbiome on the way to cancer. J. Immunol. Res. 2019, 2019, 1394191.

Chow, S. C. et al. Gram negative bacteria increase non-small cell lung cancer metastasis via Toll-like receptor 4 activation and mitogen-activated protein kinase phosphorylation. Int J Cancer 136, 1341–1350, https://doi.org/10.1002/ijc.29111 (2015).

The suggested literature is of much interest for our data and we have put some of the above-mentioned publications into the context of our data. 

Minor corrections:

6. In Pg 3 line 56 cephalosporine spelling correction to Cephalosporin

We have changed this accordingly.

7. In Pg 4 Line 83 says '(definition see below)' – no definition given

We have added a reference to the section “screening procedures and definitions” where the requested definitions can be found.

8. In Pg 4 Line 84 ‘…were included into this analysis…’ consider changing the word 'into' to 'to'.

This was corrected.

9. In Pg 4 Line 85 ‘….second malignancy - aside of localized non-melanoma skin cancer….’ consider replacing ‘aside of’ by ‘aside from’

We have changed this accordingly.

10. In Pg 5 Line 100 ‘…….unit as well as all patients all patients admitted to the thoracic surgery…..’, repetition of ‘all

patients’.

This was corrected.

11. In Pg 14, Lines 289-290, ‘We encountered a significantly higher co-occurence of diabetes in patients’ spelling of occurrence.

This was corrected.

Reviewer #2: I would like to congratulate the authors for conducting the study. It is a well written protocol. There are no previous studies on MDRO and lung cancer. The authors have clearly discussed the limitations.

Reviewer #3: The authors aimed to reveal that the impact of MDRO colonization in patients who have been diagnosed with Non-small cell lung cancer (NSCLC) who are at known high-risk for invasive infections. However, the data is not accurate enough. The data in this manuscript do not support the conclusion.

Reviewer #4: Authors have presented a very important topic on antimicrobial resistance in patients with NSCLC. Antibiotic resistance is a global problem which need be to tackled worldwide. Despite low prevalence of MDRO in their study, however this data from developed world merit to be shared to scientific community at large.

Overall the manuscript is well written and interesting to read. However, results and discussion sections need to revised. In results section some tables are difficult to understand and statistical interpretations are not well understood.

We thank the reviewer for the substantial time and effort and appreciate the kind comments and helpful suggestions.

Specific comment

Abstract:

Line 38: Replace 295 with Two hundred and ninety-five.

This has been replaced.

Line 44: Add crude and adjusted, plus their 95%CI for both univariate and multivariable analysis, respectively.

Line 43 and 44 report the Kaplan-Meier estimate of median overall survival (OS) in the MDROpos and MDROneg study groups with 95% CI. The Kaplan-Meier estimates of median survival itself or its associated CI is independent of covariables. We used multivariable Cox regression technique to adjust for additional, potentially confounding covariables in testing the association between MDRO colonization and OS (as expressed by hazard ratios and their CIs in Tab. 2). We report p values for the test of association between MDRO colonization and survival for both univariate and multivariate statistics. The multivariable Cox proportional hazards regression models would allow to estimate the probability of median survival in fixed subgroups of the data, e.g. for female MDROpos patients, or in the group of MDROpos and MDROneg patients while keeping all the other covariates fixed at their mean value. We are not aware that this is commonly done for the reporting of median survival times in trial populations. 

Line 44 – 46: “There was a significantly higher rate of non-cancer-related mortality in MDROpos compared to MDROneg patients (p<0.001) with a trend MDROpos towards an increased rate of fatal infections in patients (p=0.05)” from table 3 I found very difficult to interpret. The way p-value which has been presented it looks to reject null hypothesis in favor of alternative hypothesis. Considering NRM cases alone in comparison of non-cancer related mortality might obscure the intended outcome, if possible include all participants in this analysis. Table 3 need to be revised for more clarity.

We agree with the reviewer that Table 3 is difficult to interpret. We have therefore revised Table 3 to display the data in a more intuitive way. The table legend has been rewritten to describe the statistical testing carried out in more detail in order to allow interpretation of the p value. Additionally, we have included a comment on the statistical test performed.

The null hypothesis being tested here was an equal distribution of infectious-related deaths in the MDROpos and MDROneg subgroups. We hypothesized that infectious-related deaths might account for the increased non-relapse mortality observed in MDROneg patients and therefore tested this. The death of the other participants was either due to relapse or progression or due to unknown/indeterminate causes. We agree with the notion that infection-related complications might also be a clinically relevant factor in relapsed/progressed patients and that an analysis of the distribution of infections between these patients might give additional information. However, we relied on the cause of death as described in the final medical report or on the death certificate. We unfortunately do not have sufficient information on the final course of each individual patient to judge the contribution of infectious-related complications to the death of patients with progressive cancer. However, we believe that information gained from this exploratory analysis is rather hypothesis generating than convincing evidence, we therefore weakened our claims regarding this analysis throughout the manuscript. 

Introduction:

Line 55–56: Third-generation cephalosporin resistant should be replaced with extended spectrum beta lactamase producers.

We thank the reviewer for this suggestion. After careful consideration, we have opted not to make this change, since “third-generation cephalosporin resistant” also encompasses cephalosporine resistance which is not due to ESBLs but instead due to other cephalosporinases such as AmpC beta-lactamases in Enterobacter, Serratia and others (see e.g. DOI 10.1128/CMR.00036-08). 

Line 57: Replace Staphylococcus aureus with methicillin resistance to Methicillin resistance Staphylococcus aureus

We have changed this accordingly.

Materials and Methods

ESBL, MRSA and VRE were screened by screening media, these media have high sensitivity. It will be interesting to state how this MDRO were confirmed either by phenotypic or genotypic.

As pointed out in the chapter “detection and molecular resistance analysis in MDRO”, VRE, ESBL and MRSA species are confirmed PHENOTYPICALLY by CLSI certified VITEK and/or agar diffusion method. 

Line 146: Ensure consistency either use MDROpos or MDRO+ .

We have corrected this.

Results

Line 168: Replace 271 with Two hundred and seventy-one.

We have corrected this.

Line 173: Diabetes mellitus did not show significant association, revisit supplementary file OR included 1.

Table 1 shows that Diabetes mellitus was present in 50% of patients with MDRO colonization and 16% of MDROneg patients, and that this association was highly significant. 

Table1: Some variable in a column total % do not add to 100%. For example, in co-morbidity variable total number of MDRO is 25 and not 24. Interpretation along the row could be interesting than along the column. Think of revising this table.

Comorbidites has non-mutually exclusive categories (some patients had multiple comorbidities), explaining while the patient numbers do not sum to 100%. However, while checking Table 1 we noticed that the entry “1st line treatment approach” was missing information for 5 patients. We have corrected this and apologize for the mistake. 

Line 197 – 207: Specify name of specific antibiotics tested instead of using classes of antibiotics. For examples macrolide use either erythromycin or azithromycin, aminoglycosides specify gentamicin or others.

We have added this data.

In analysis MDRO+ exposure to OS outcome. The disease stage could be one of the important confounders for OS. See Fig1A for example, for interest I would like to know if you controlled for disease stage as confounder, what happen HR. If you did not do please could you explain to me for interest.

We agree with the reviewer that disease stage is a potential confounder for OS. We have therefore already adjusted for this in our multivariate Cox proportional hazards regression analysis of overall survival (see Tab. 2, limited vs extensive disease as well as Fig. 2). As expected, extensive disease was highly significantly associated with a poorer OS (HR 3.03, 95% CI 2.16–4.24). The association between MDRO colonization and OS remained significant in this multivariable analysis in the overall cohort (Tab. 2). 

Although this has been explained in discussion, however authors need to dig further on MDRO colonization with inferior overall survival outcome. It is well hypothesized MDRO colonization is a risk for severe MDRO infection with the same bacteria. MDRO infection could have direct effect on overall survival in these group of patients.

We agree with the reviewer that MDRO colonization is a likely risk factor for infection by the colonizing bacteria. In support of this hypothesis, we have therefore statistically tested the rate of mortality due to infections in MDROpos vs. MDROneg patients and observed statistical significance (p=0.002) towards an increased rate of infection in MDROpos patients (see revised Table 3; this relates to the reviewer’s remark and our answer above in regard to Table 3.). At the end of the results section (‘Cause of death’), we further discuss all MRDOpos patients within our cohort which died due to complications relating to infections and also provide information regarding the colonizing pathogens later detected in invasive infections, as well as discuss the infections. 

Analysis on Fig2B and 2C could be combined (IV/IIIB vs IIIA/II/I) rather than categorized in different group. Since disease staging in itself could predict OS.

We agree with the reviewer that disease stage should be part of the full survival analysis, since it likely impacts survival. As outlined in our answer above, we have already included the disease stage as a variable in the multivariable survival analysis. We have included the separate (IV/IIIB vs IIIA/II/I) Kaplan-Meier OS estimates in Fig. 2B and 2C because we believe it is of clinical interest to depict the associative impact of MDRO colonization in these two stages. However, our primary outcome analysis (multivariable survival analysis) is performed with the full patient cohort, taking into account the disease stage for each patient. The association between MDRO colonization and OS remained significant in the multivariable analysis of the overall cohort (HR 1.96, 95% CI 1.09-3.51; see Tab. 2). 

Line 246 – 251 looks like figure 2 legend, please move below the figure2

We totally agree with the reviewer that placing the Figure´s legends into the running text (and not under the corresponding Figure) is confusing, unfortunately this is part of the PlosOne author guidelines, so we are not able to place the legend(s) below the corresponding figures.

Line 256 – 258: Check comment on abstract.

Table 3: Check comment on abstract

As outlined in our answer above, we have revised Table 3 and our discussion of it.

Line 267 – 268: Looks like legend for figure 3.

Please see above.

Discussion:

Well written, however in some part authors need to revise like line 302. If table 3 is revised this statement might need to change, there is a need to include all mortality not NRM only.

In conclusion, line 350 authors did not establish the correlation between MDRO colonization and infection. To state this as a reason “due higher rate of fatal infections mostly involving the lung tissue”. This infection was unrelated to colonization, the statement needs to be revised.

We agree with the reviewer and have revised this section. No claim of causality is made any more in this regard. While suggestive, our retrospective trial design does not allow us to identify causal factors. In order to address this justified criticism, we mitigated our claims suggestive of causality throughout the manuscript. However, this does not pertain to our main finding of significantly poorer overall survival in MDROpos NSCLC patients.

---

## [Decision Letter · Decision Letter 1]

3 Nov 2020

PONE-D-20-09321R1

Colonization with multi-drug-resistant organisms negatively impacts survival in patients with non-small cell lung cancer

PLOS ONE

Dear Dr. Stratmann,

Thank you for submitting your manuscript to PLOS ONE. After careful consideration, we feel that it has merit but does not fully meet PLOS ONE’s publication criteria as it currently stands. Therefore, we invite you to submit a revised version of the manuscript that addresses the points raised during the review process.

We look forward to receiving your revised manuscript.

Kind regards,

Joel Manyahi

Academic Editor

PLOS ONE

Reviewers' comments:

Reviewer's Responses to Questions

**Comments to the Author**

1. If the authors have adequately addressed your comments raised in a previous round of review and you feel that this manuscript is now acceptable for publication, you may indicate that here to bypass the “Comments to the Author” section, enter your conflict of interest statement in the “Confidential to Editor” section, and submit your "Accept" recommendation.

Reviewer #1: All comments have been addressed

Reviewer #2: All comments have been addressed

2. Is the manuscript technically sound, and do the data support the conclusions?

Reviewer #1: Yes

Reviewer #2: Yes

3. Has the statistical analysis been performed appropriately and rigorously? 

Reviewer #1: Yes

Reviewer #2: Yes

4. Have the authors made all data underlying the findings in their manuscript fully available?

Reviewer #1: Yes

Reviewer #2: Yes

5. Is the manuscript presented in an intelligible fashion and written in standard English?

Reviewer #1: Yes

Reviewer #2: (No Response)

6. Review Comments to the Author

Reviewer #1: The authors have considered all suggestions and corrections addressed by the reviewer. Comments have been added and additional references have been cited as suggested. Minor corrections have also been addressed. The most outstanding limitations were acknowledged. This makes the manuscript more acceptable from a readers point of view.

However, the author needs to correct one spelling mistake, that is 'thirdly' in line 326. Otherwise, the manuscript is acceptable.

Reviewer #2: The authors have answered all queries. I have no further comments, The manuscript will be useful for the readers.

7. PLOS authors have the option to publish the peer review history of their article (what does this mean?). If published, this will include your full peer review and any attached files.

Reviewer #1: **Yes: **Dr. Sunjukta Ahsan, Department of Microbiology, University of Dhaka, Bangladesh

Reviewer #2: No

---

## [Author Response · Author response to Decision Letter 1]

3 Nov 2020

Reviewer #1: The authors have considered all suggestions and corrections addressed by the reviewer. Comments have been added and additional references have been cited as suggested. Minor corrections have also been addressed. The most outstanding limitations were acknowledged. This makes the manuscript more acceptable from a readers point of view.

However, the author needs to correct one spelling mistake, that is 'thirdly' in line 326. Otherwise, the manuscript is acceptable.

We thank the referee again for his time to review our revised manuscript and have applied the last suggested change.

Reviewer #2: The authors have answered all queries. I have no further comments, The manuscript will be useful for the readers.

We thank the reviewer for his valuable time.

---

## [Editor Report · Decision Letter 2]

5 Nov 2020

Colonization with multi-drug-resistant organisms negatively impacts survival in patients with non-small cell lung cancer

PONE-D-20-09321R2

Dear Dr. Stratmann,

We’re pleased to inform you that your manuscript has been judged scientifically suitable for publication and will be formally accepted for publication once it meets all outstanding technical requirements.

Kind regards,

Joel Manyahi

Guest Editor

PLOS ONE
---

## [Editor Report · Acceptance letter]

12 Nov 2020

PONE-D-20-09321R2 

Colonization with multi-drug-resistant organisms negatively impacts survival in patients with non-small cell lung cancer 

Dear Dr. Stratmann:

I'm pleased to inform you that your manuscript has been deemed suitable for publication in PLOS ONE. Congratulations! Your manuscript is now with our production department. 

Kind regards, 

on behalf of

Dr. Joel Manyahi 

Guest Editor

PLOS ONE